# The Ameliorative Effect of *Litsea martabanica* (Kurz) Hook. f. Leaf Water Extract on Chlorpyrifos-Induced Toxicity in Rats and Its Antioxidant Potentials

**DOI:** 10.3390/foods13111695

**Published:** 2024-05-28

**Authors:** Weerakit Taychaworaditsakul, Suphunwadee Sawong, Supaporn Intatham, Sunee Chansakaow, Phraepakaporn Kunnaja, Teera Chewonarin, Kanjana Jaijoy, Absorn Wittayapraparat, Pedcharada Yusuk, Seewaboon Sireeratawong

**Affiliations:** 1Clinical Research Center for Food and Herbal Product Trials and Development (CR-FAH), Faculty of Medicine, Chiang Mai University, Chiang Mai 50200, Thailand; subhawat.s@cmu.ac.th (W.T.); suphunwadee.sa@cmu.ac.th (S.S.); intatham_s@outlook.com (S.I.); 2Department of Pharmacology, Faculty of Medicine, Chiang Mai University, Chiang Mai 50200, Thailand; 3Department of Biochemistry, Faculty of Medicine, Chiang Mai University, Chiang Mai 50200, Thailand; teera.c@cmu.ac.th; 4Department of Pharmaceutical Sciences, Faculty of Pharmacy, Chiang Mai University, Chiang Mai 50200, Thailand; sunee.c@cmu.ac.th; 5Department of Medical Technology, Faculty of Associated Medical Sciences, Chiang Mai University, Chiang Mai 50200, Thailand; phraepakaporn.k@cmu.ac.th; 6McCormick Faculty of Nursing, Payap University, Chiang Mai 50000, Thailand; joi.kanjana@gmail.com; 7Highland Research and Development Institute (Public Organization), Chiang Mai 50200, Thailand; absornw@hrdi.or.th (A.W.); npedcharada@gmail.com (P.Y.)

**Keywords:** *L. martabanica* leaf, water extraction, antioxidant, anti-pesticide, medicinal plant, acetylcholinesterase activity, hepatoprotective, access to healthcare

## Abstract

*Litsea martabanica* root’s antioxidant and acetylcholinesterase (AChE) activity showed promise as a pesticide detoxification agent in our previous study. In addition to its root, leaves can help alleviate pesticide exposure, although there is limited scientific evidence supporting their efficacy. However, the use of roots in several countries, such as Thailand, could contribute to environmental degradation, as highland communities traditionally used leaves instead of roots. This study aims to evaluate the antioxidant activity and anti-pesticide potential of water extract from *L. martabanica* leaves through in vitro and in vivo investigations. In the in vitro study, *L. martabanica* water extract and its fractions demonstrated antioxidant activity and induced apoptosis in hepatic satellite cells. In the in vivo study, treatment with the leaf extract led to increased AChE activity, decreased malondialdehyde (MDA) levels, increased superoxide dismutase (SOD) levels, and reduced glutathione in chlorpyrifos-exposed rats. Histopathological examination revealed that chlorpyrifos-treated rats exhibited liver cell damage, while treatment with the water extract of *L. martabanica* exhibited a protective effect on the liver. In conclusion, *L. martabanica* water extract exhibited antioxidant activity, enhanced AChE activity, and improved histopathological abnormalities in the liver.

## 1. Introduction

Pesticides play a crucial role in agriculture globally, where they are extensively employed to enhance and protect crop yields by controlling insects, weeds, mollusks, and fungi [1]. However, the long-term use of chemical pesticides has been shown to have adverse effects on ecosystems. There is also increasing evidence of negative impacts on human health, including acute neurologic poisoning, chronic neurodevelopmental impairment [2], cancers [3], liver diseases [4], and potential immune dysfunction [5]. Most pesticides, such as chlorpyrifos, are synthetic compounds consisting primarily of organophosphates, which irreversibly inhibit the synthesis of acetylcholinesterase (AChE) [6,7]. In addition, pesticides can result in oxidative damage by increasing lipid peroxidation, such as malondialdehyde (MDA), hydrogen peroxide (H_2_O_2_), superoxide anion (O_2_^•−^), reactive nitrogen species (RNS), and reducing the concentrations of reduced glutathione, a non-enzymatic antioxidant, as well as antioxidant enzymes, such as catalase, superoxide dismutase, and glutathione peroxidase [8,9], in various organs, notably the liver [10,11]. Increased lipid peroxidation and decreased AChE activity are consequences of occupational pesticide exposure in humans [12,13]. These conditions result in excessive stimulation of both muscarinic and nicotinic receptors, leading to symptoms such as cramps, lacrimation, paralysis, muscular weakness, muscular fasciculation, diarrhea, and blurred vision [14,15,16]. These symptoms can begin within minutes or hours of exposure and can persist for days to weeks. Additionally, continual exposure to pesticides has been linked with the onset of liver disorders, such as disrupted hepato-biliary functions, liver injury, hepatitis, and fibrosis [17,18,19]. Importantly, the activation of hepatic stellate cells (HSCs) plays a pivotal role in the formation of liver fibrosis. Strategies aimed at inhibiting hepatic stellate cell proliferation and promoting apoptosis hold promise in the treatment of fibrosis [20]. Moreover, antioxidants have been proposed as a potential treatment for acute organophosphate pesticide poisoning because of their ability to counteract ROS damage in the liver [21,22].

The *Lauraceae* family encompasses 52 genera, totaling around 2550 species, which are distributed primarily in tropical and warm regions of southeastern Asia and Brazil [23]. Among these, *Litsea martabanica* (Kurz) Hook. f. is found in Thailand, China, and Myanmar [24]. The traditional utilization of this plant is deeply ingrained in the traditional wisdom of highland communities. Various parts of the plant, including roots, leaves, and stems, have been used as medicinal remedies in the highland areas of northern Thailand. These applications extend to treating kidney disease, mitigating toxic allergy symptoms, and facilitating detoxification [25]. Plants are recognized as valuable resources for mitigating reactive oxygen species (ROS), potentially serving as food ingredients to counteract the adverse effects of pesticides [26,27]. An in vivo experimental study found that green tea extracts, abundant in polyphenols, notably decreased pesticide-induced oxidative stress and pulmonary fibrosis [28]. Olive leaf extract has also been shown to have notable antioxidant and antiapoptotic effects in rats exposed to chlorpyrifos-induced neurotoxicity and reproductive toxicity [29]. In a previous study, we highlighted the root of *L. martabanica* for its potential as an anti-pesticide detoxifying agent, which was attributed to its substantial antioxidant and AChE activity [17]. That investigation also discovered, drawing on the wisdom of highland communities, that in addition to boiling the plant’s roots for consumption to mitigate pesticide exposure, the leaves can also be utilized for the same purpose.

The present study aims to investigate the *L. martabanica* leaf water extract using in vitro and in vivo testing to gather scientific data on its potential use as an anti-pesticide. Additionally, this study assesses the impacts of the leaf water extract from *L. martabanica*, including the desirable properties described in traditional knowledge and wisdom, for developing standard products.

## 2. Materials and Methods

### 2.1. Reagents

Dulbecco’s Modified Eagle Medium-high glucose (DMEM-HG; 12800-58), fetal bovine serum (FBS), phosphate-buffered saline (PBS), and trypsin-EDTA solution were purchased from Gibco (Grand Island, NY, USA). Chlorpyrifos, dimethyl sulfoxide (DMSO), and sulforhodamine B (SRB) were purchased from Sigma Chemical, Inc. (St. Louis, MO, USA). Penicillin, streptomycin, and L-glutamine were obtained from Invitrogen (Thermo Fisher Scientific Inc., Waltham, MA, USA).

### 2.2. Plant Material and Quality Control of Raw Materials

*Litsea martabanica* was gathered from Chiang Mai Province, Thailand, in June 2020, and identified by a taxonomist. A voucher specimen (No. WP 7185) was deposited at the Queen Sirikit Botanical Garden. The leaves were chosen, cut into small pieces, dried in a hot-air oven until the moisture content was below 10%, and then pulverized. The leaf powder of *L. martabanica* was evaluated for the quality of the raw material, including loss on drying, ash values (total ash and acid-insoluble ash), and extractive values (ethanol-soluble and water-soluble extractive values), following the methods described in the Thai Herbal Pharmacopoeia 2018 [30].

### 2.3. Extract Preparation

The powder of *L. martabanica* leaves was extracted by decoction using water as a solvent, with a ratio of 1 L of water to 100 g of leaves (soaked for 30 min) over 1.5 h. The filtrate was obtained by filtering the extract through filter paper, and then the marc of plant material was re-extracted twice. Subsequently, the filtrate was gathered and concentrated to achieve a concentration of approximately 3% Brix, and the acid–base balance was measured. The extract was dried using a spray dryer (BUCHI Mini Spray Dryer B-290) at an inlet temperature of 140 °C and an outflow temperature of 85 ± 5 °C, with the aspirator set at 100% and the pump at 25%, resulting in a water-soluble powder.

### 2.4. Chemical Profile by Thin-Layer Chromatography

The water extract of *L. martabanica* leaves (5 mg) was dissolved in 1 mL of aqueous ethanol to prepare the test solution. Standards of apigenin, caffeic acid, ellagic acid, gallic acid, kaempferol, quercetin, and rutin were each prepared (concentration of 1 mg/mL). A CAMAG HPTLC system from Muttenz, Switzerland, was utilized, consisting of a Linomat 5 automatic applicator with a 10 mL syringe, a CAMAG Automatic Developing Chamber 2 (ADC 2), a CAMAG TLC scanner 4, and winCATS software version 1.4. For HPTLC fingerprinting analysis, 2 µL each of the test solution and standard solutions were applied as 8 mm-long bands on a Silica Gel GF254 TLC plate. The plate was then placed in a TLC twin-trough developing chamber, pre-saturated with solvent vapor, using a mobile phase consisting of ethyl acetate, methanol, and water in the ratio of 80:20:2. Densitometric scanning was performed using a TLC scanner with winCATS software at 254, 280, and 320 nm. The plate was placed in a photo-documentation chamber (CAMAG TLC Visualizer 2) and images were captured under white light, UV 254 nm, and UV 366 nm. The developed plate was then sprayed with anisaldehyde, natural product, and DPPH spraying reagents, and dried at 100 °C in a hot-air oven.

### 2.5. DPPH Assay

The antioxidant capacity of *L. martabanica* was assessed using the DPPH (2,2-diphenyl-1-picrylhydrazyl) radical scavenging assay [31]. In that process, DPPH was dissolved in methanol to achieve a final concentration of 80 µg/mL. Subsequently, the extracts, dissolved in DMSO, underwent dilution to varying concentrations (25–200 µg/mL). Each diluted extract (20 µL) was then dispensed into separate wells. Following this, 180 µL of DPPH solution was added to each well. The plate was then left to incubate at room temperature in darkness for 30 min. Absorbance readings were taken at 517 nm using a microplate reader. Vitamin C and methanol served as the reference standard and control, respectively. The percentage of DPPH scavenging activity was determined, and the concentration of the sample necessary to inhibit 50% of DPPH radicals was quantified as the IC_50_ value.

### 2.6. Superoxide Radical Assay

An assay was conducted to assess the antioxidant efficacy of the test sample against superoxide free radicals. Superoxide radicals were generated through the action of phenazine methosulfate (PMS) and nicotinamide adenine dinucleotide (NADH), subsequently reducing nitroblue tetrazolium (NBT) to yield purple formazan [32,33]. Reagents, including PMS (25 µM), NADH (0.5 mM), and NBT (0.2 mM), were prepared by dissolving them in a phosphate buffer solution (pH 7.4). The experimental procedure involved adding 50 µL each of NBT solution, NADH solution, plus varying concentrations (25–200 µg/mL) of the test sample to individual wells of a 96-well plate, followed by the addition of 50 µL of PMS solution. After a 10 min incubation at room temperature, the optical density (OD) was measured at 560 nm using a microplate reader. Gallic acid and PBS solutions were used as the reference standard and control, respectively. Subsequently, the percentage of superoxide radical scavenging and IC_50_ values were determined using the same formula as the DPPH assay [17].

### 2.7. Cell Culture

Human hepatic stellate cell line (LX-2) was purchased from Merck Millipore (Burlington, MA, USA) and cultured in Dulbecco’s Modified Eagle Medium with 100 Units/mL of penicillin and 100 µg/mL of streptomycin, supplemented with 2% heat-inactivated fetal bovine serum and 2 mM L-Glu. The cells were incubated at 37 °C in a 5% CO_2_ atmosphere. When the cells reached a confluence of 70–80%, they were harvested, plated, or sub-cultured for preservation or for use in subsequent passages.

### 2.8. Cell Viability Assay

The SRB assay procedure commenced with the seeding of cells at a density of 10,000 cells per well in a 96-well plate, each well containing 100 µL of complete medium. Following this, the extract treatments, also prepared in 100 µL of complete medium, were administered and incubated for either 24 or 48 h at 37 °C under 5% CO_2_ conditions. Subsequently, cell viability was evaluated utilizing the SRB assay, with comparisons made against untreated cells. Post-treatment, 40 µL of 50% trichloroacetic acid (TCA) was added to each well and the plates were incubated at 4 °C for one hour, after which the plates underwent four washes with tap water and were subsequently dried. Following that, 100 µL of 0.057% SRB solution was added to each well for a 30 min incubation period, followed by four washes with 1% acetic acid to eliminate unbound dye. After drying, 200 µL of 10 mM Tris-based solution (pH 10.5) was introduced, and the absorbance at 510 nm was measured using a microplate reader (BioTek, Winooski, VT, USA) [34].

### 2.9. Apoptosis Assay by Flow Cytometry

The LX-2 cells were plated at a density of 3 × 10^5^ cells in 6-well plates and exposed to each extract, ranging from 25 to 100 µg/mL, for 24 h. Following treatment, the cells were collected and washed twice with PBS, after which they were stained with 50 µL of binding buffer containing annexin V-FITC and propidium iodide (PI) for 15 min. Following this, 250 µL of binding buffer was added to the stained cells, and the samples were analyzed using a BD FACScan™ flow cytometer (BD Biosciences, San Jose, CA, USA). The results were evaluated based on the percentage of cells exhibiting positive signals for annexin V-FITC and PI, or only annexin V-FITC [35].

### 2.10. Anti-Pesticide Potential

#### 2.10.1. Animals

Male Sprague-Dawley rats, weighing 180–200 g, were obtained from Nomura Siam International (Nomura Siam International Co., Ltd., Bangkok, Thailand). Standard environmental conditions were adhered to, including a temperature of 24 ± 1 °C and a 12 h dark–light cycle for their housing. All animals were provided with unrestricted access to drinking water and a regular pellet diet. They were acclimated for a minimum of one week before commencing the studies. The Animal Ethics Committee of the Faculty of Medicine, Chiang Mai University, approved all experimental protocols (approval number 49/2559).

#### 2.10.2. Experimental Groups

The anti-pesticide potential of *L. martabanica* leaf water extract was assessed using a previously reported method [17]. Male rats were divided into five groups of six animals each. Group 1 (normal group) was orally administered 2 mL/kg of distilled water daily for 16 days. Group 2 (control group) received the same dose of distilled water plus chlorpyrifos (Sigma Chemical, Inc., St. Louis, MO, USA) at a dosage of 16 mg/kg daily for the same 16-day period. The rats in groups 3 to 5 were orally administered varying doses of *L. martabanica* leaf water extract 30 min before receiving chlorpyrifos at a dosage of 16 mg/kg daily for 16 days. The administration of the leaf extract acted as an imitation of the concoction based on traditional knowledge used in highland communities. In group 3, rats were subjected to a cyclic dosing regimen of *L. martabanica* leaf water extract, receiving a cyclical daily administration of 7.5 mg/kg for 2 days followed by 2.5 mg/kg for 2 days over a period of 16 days (four rounds). Group 4 received a cyclic dose of 75 mg/kg for 2 days followed by 25 mg/kg for 2 days, and group 5 received a cyclic dose of 750 mg/kg for 2 days followed by 250 mg/kg for 2 days, also with daily administration over 16 days (4 rounds). Behavioral changes following the administration of chlorpyrifos and *L. martabanica* leaf water extract were observed in the rats. Signs of toxicity, especially nicotinic- (muscular fasciculation and weakness) and muscarinic-related signs (small pupils, tremors, diarrhea, salivation, lacrimation, and others), were observed and recorded. Additionally, the rats’ body weights were measured once daily throughout the experiment.

Following the end of the period of daily administration, on the 17th day, all rats were anesthetized via an intraperitoneal injection of 120 mg/kg of thiopental sodium. Vital signs, pulse, and reflexes were assessed to confirm the animals’ death. Blood samples were obtained for hematological and biochemical analysis. The liver and kidneys were removed for weighing, gross pathology, and histopathological examination.

#### 2.10.3. Assay of AChE Activity

An AChE activity kit was used following the manufacturer’s instructions (Sigma Chemical, Inc., St. Louis, MO, USA). Briefly, 10 µL of diluted whole blood samples were dispensed into each well of a 96-well plate, followed by the addition of 190 µL of the working reagent to all wells. The reaction mixtures were then left to incubate at room temperature, and absorbance readings were recorded after 2 and 10 min using a microplate reader set at 412 nm [17].

#### 2.10.4. Assay of Malondialdehyde (MDA), Superoxide Dismutase (SOD), and Reduced Glutathione

The serum was then analyzed for malondialdehyde (MDA) levels using the enzyme-linked immunosorbent assay (ELISA) kit from Elabscience (Houston, TX, USA), for superoxide dismutase (SOD) levels using the RANSOD kit (Randox Lab, Crumlin, UK), and for reduced glutathione levels using the Sigma Aldrich Glutathione Assay kit (Millipore Sigma, Burlington, MA, USA; Cat. No. MAK364), following the manufacturer’s protocol.

#### 2.10.5. Hematological and Biochemical Analysis

For hematological analysis, whole blood was collected in a tube containing ethylenediaminetetraacetic acid (EDTA) and subjected to measurement using a Mindray BC-5300 Vet automated hematology analyzer (Shenzhen, China). For biochemical analysis, blood was collected in a clot-activated tube. Following centrifugation at 3500 rpm for 10 min, the blood biochemical parameters were determined using an automated BX-3010 analyzer (Sysmex, Kobe, Japan) provided by the Small Animal Hospital at Chiang Mai University.

#### 2.10.6. Histopathology Examination

The liver and kidney were preserved in 10% formalin and subjected to gradient ethanol dehydration and paraffin embedding. The paraffin blocks were sliced into sections (thickness, 4 μm) and stained with hematoxylin and eosin (H&E). The histopathological characteristics were observed under an optical microscope (Olympus CX-23; Olympus Corporation, Tokyo, Japan).

### 2.11. Statistical Analysis

Results are expressed as mean ± S.E.M. The initial analysis involved the Shapiro–Wilk test to check for normality. If the data showed no significant deviation from a normal distribution, ANOVA followed by Tukey’s multiple comparison tests were conducted. Conversely, if the Shapiro–Wilk test indicated a non-normal distribution, the Kruskal–Wallis nonparametric ANOVA test followed by Dunn’s test were applied. Statistical significance was set at *p* < 0.05. Statistical analyses were performed using IBM SPSS Statistics, version 22.0 (International Business Machines Corporation, Armonk, NY, USA).

## 3. Results

### 3.1. Quality Control of Raw Materials and Extracts

The physicochemical examinations, loss on drying, total ash, acid-insoluble ash, ethanol-soluble extractive value, and water-soluble extractive value of the raw material are presented in Table 1, along with their mean values.

The leaves of *L. martabanica* were extracted using traditional methods that involved decoction using water as the solvent. Although the leaf of *L. martabanica* is not included in any pharmacopeia or textbook, this study developed the specification of *L. martabanica*’s leaf alignment following the Thai Herbal Pharmacopeia methods for quality control when handling raw materials, which could assist future research.

### 3.2. Chemical Profile by Thin-Layer Chromatography

Chemical profiles of the water extract of *L. martabanica* leaf were determined by HPTLC with the use of a detector under UV light at 254 and 366 nm. The plates were sprayed with a freshly prepared anisaldehyde-sulfuric acid reagent, DPPH, or natural product spraying reagent. The major component was determined at Rf 0.46, 0.58, 0.66, and 0.80, at 254 nm. Based on comparison with chemical standards, the extract of *L. martabanica* was found not to contain apigenin (Rf 0.91), caffeic acid (Rf 0.77), ellagic acid (Rf 0.03), gallic acid (Rf 0.75), kaempferol (Rf 0.91), quercetin (Rf 0.88), or rutin (Rf 0.23).

### 3.3. Antioxidant Activity by DPPH Assay and Superoxide Radical Scavenging Activity

The half-maximal inhibitory concentration (IC_50_) values for both DPPH and superoxide free radical scavenging activity are presented in Table 2. Among these fractions, the ethyl acetate fraction demonstrated greater antioxidant properties compared to both other fractions and the crude extract. Additionally, the gallic acid (reference standard) exhibited the highest ability in scavenging free radicals. Both DPPH and superoxide radical assays revealed that the IC_50_ of each extract was significantly different from that of gallic acid.

### 3.4. Cytotoxic Effects of L. martabanica Leaf Water Extract and Its Fractions on Human Hepatic Stellate Cells

To evaluate the impact on cell viability, various concentrations (12.5–800 µg/mL) of water extract from *L. martabanica* leaf and its fractions were prepared, and then incubated with LX-2 cells for 24 and 48 h, as shown in Figure 1. The IC_50_ value of *L. martabanica* leaf water extract exceeded 1000 µg/mL and was 564.70 ± 40.95 µg/mL at 24 and 48 h, as depicted in Table 3. Interestingly, the ethyl acetate fraction exhibited toxicity toward LX-2 cells, with IC_50_ values of 154.7 ± 11.29 µg/mL at 24 h and 539.90 ± 38.26 µg/mL at 48 h (Table 3). Other fractions, including butanol and residue, showed relatively low toxicity, with IC_50_ values exceeding 500 µg/mL at both 24 and 48 h.

To confirm apoptosis induction by *L. martabanica* leaf water extract, apoptosis cell death was assessed using annexin V-fluorescein isothiocyanate (FITC)/propidium iodide (PI) staining and analyzed through flow cytometry. The results revealed that *L. martabanica* leaf water extract tended to induce apoptosis in LX-2 cells. Interestingly, ethyl acetate at a high concentration (800 µg/mL) was capable of significantly inducing apoptosis, while butanol and residue extracts did so at concentrations of 400 and 800 µg/mL, respectively (Figure 2).

### 3.5. Anti-Pesticide Potential in Rats

#### 3.5.1. Acetylcholinesterase Activity

The normal group of rats showed an acetylcholinesterase activity value of 2008 ± 182 U/L. In contrast, rats in the control group treated with chlorpyrifos exhibited significantly decreased average acetylcholinesterase activity levels compared to normal rats. Conversely, rats receiving *L. martabanica* leaf water extract at all concentrations displayed significantly increased acetylcholinesterase activity levels compared to the control group that received only chlorpyrifos (Figure 3A).

#### 3.5.2. Malondialdehyde (MDA), Superoxide Dismutase (SOD), and Reduced Glutathione in Serum

Regarding serum MDA levels, rats treated with chlorpyrifos showed a tendency for increased levels (0.057 ± 0.005 nmol/mg protein), though not significantly different from the normal rat group. However, rats receiving *L. martabanica* leaf water extract at all concentrations, along with chlorpyrifos, exhibited significantly decreased MDA levels compared to the control group that received chlorpyrifos alone (Figure 3B).

The experiment also revealed elevated SOD levels in the serum of rats treated with chlorpyrifos to be statistically significantly different from the normal group. Conversely, rats receiving *L. martabanica* leaf water extract at all doses, along with chlorpyrifos, showed a statistically significant increase in SOD enzyme levels compared to the control group that received only chlorpyrifos (Figure 3C). Regarding reduced glutathione levels in serum, the control group receiving chlorpyrifos displayed a level of 0.159 ± 0.010 nmol/mg protein, which was significantly different from the normal group. Rats receiving medium doses of *L. martabanica* leaf water extract (75 and 25 mg/kg), along with chlorpyrifos, showed a tendency toward increased reduced glutathione levels, although the values were not statistically significantly different. However, rats receiving high doses of *L. martabanica* leaf water extract (750 and 250 mg/kg), along with chlorpyrifos, exhibited significantly higher levels of reduced glutathione in serum than the chlorpyrifos-treated group (*p* < 0.05), with values similar to those in normal rats receiving distilled water (Figure 3D).

#### 3.5.3. Body Weight and Organ Weight Change

This experiment involved administering the insecticide chlorpyrifos at a dose of 16 mg/kg body weight daily for 16 days to rats in all three groups receiving *L. martabanica* leaf water extract. The results revealed a higher body weight in rats from the group receiving low and middle doses of the extracts combined with the insecticide compared to that of the control group on the first day, as depicted in Table 4. There were no statistically significant alterations in the weight of the liver. Additionally, the kidney weight of only the group receiving the extract combined with the insecticide significantly increased compared to the normal and control groups.

#### 3.5.4. Hematological Analysis

In the hematological analysis investigating the anti-pesticide effect, it was found that the rats in the groups receiving *L. martabanica* leaf water extract in middle doses (75 and 25 mg/kg) and high doses (750 and 250 mg/kg) showed a statistically significant decrease in red blood cell count, hemoglobin, and hematocrit levels compared to the normal groups, as presented in Table 5. The rats in the groups receiving *L. martabanica* leaf water extract in the high doses (750 and 250 mg/kg) showed statistically significant decreases in hemoglobin and hematocrit levels compared to the control groups. The white blood cell count parameters were not significantly different from the control group (Table 6).

#### 3.5.5. Blood Chemistry Analysis

In the blood chemistry analysis to assess liver and kidney functions, as shown in Table 7, it was found that the rats receiving *L. martabanica* leaf water extract at high doses (750 and 250 mg/kg) had significantly decreased BUN and alkaline phosphatase levels compared to the normal group. Additionally, ALT levels in chlorpyrifos-treated rats tended to first increase and then revert to normal levels following administration of *L. martabanica* leaf water extract. Moreover, the control group rats receiving distilled water along with the insecticide demonstrated a statistically significant decrease in total protein levels compared to normal rats. Rats in the group receiving *L. martabanica* leaf water extract in all three conditions had a statistically significant decrease in total protein levels compared to the normal group.

#### 3.5.6. Histopathological Examination

In the histopathology examination of rats receiving chlorpyrifos (control group), dilation or widening of the sinusoids with liver hypertrophy was observed, but it was not observed in the normal group. Nevertheless, no signs of scattered cell necrosis or lymphocytic infiltration were observed. The cells displayed varying shapes and sizes, with hyperchromatic and hypertrophied nuclei (Figure 4B). Interestingly, the dilation or widening of sinusoids was notably reduced, but no hepatic necrosis was observed in the liver of rats. All concentrations of *L. martabanica* water extract could protect liver cells from chlorpyrifos-indued hepatotoxicity, especially at high concentrations (750 and 250 mg/kg; Figure 4C). For the kidneys, histopathological examination revealed normal characteristics in rats of the control group receiving distilled water and the insecticide. Leucocyte infiltration, glomerulus atrophy, and renal tubule vacuolization were not observed in rats that were exposed to chlorpyrifos. Additionally, rats that received all concentrations of the *L. martabanica* water extract, along with chlorpyrifos, did not show any abnormality in the kidney (Figure 5A–C).

## 4. Discussion

An investigation was performed to study the detoxifying and antioxidant properties of the root of *L. martabanica* in reducing the toxicity induced by chlorpyrifos in rats [17], which is consistent with the traditional approach to therapeutic use applied by Thai highland communities. Using leaves rather than roots could help reduce the environmental degradation caused by highland communities, who traditionally cut down whole trees to obtain the roots. In this study, the identification and authentication of herbal drug substances were crucial to ensuring their authenticity before proceeding to further steps. For that reason, the physicochemical examination, total ash, acid-insoluble ash, ethanol-soluble extractive value, water-soluble extractive value, and water content of the crude drugs (Table 1) will be among the criteria used to evaluate the quality of the raw material used in future studies. To that end, we examined the microscopic characteristics and chemical compositions to establish *L. martabanica*’s monograph to help ensure appropriate quality control of raw materials and extracts. Additionally, we investigated the antioxidant effects and anti-pesticide capabilities of *L. martabanica* as an aid to the further development of products for human use. This effort aligns with the project’s intention to employ traditional knowledge and wisdom and potentially transform *L. martabanica* leaf water extract into various forms, including granules, infusions, and effervescent tablets for possible wider application.

Previous research has shown that phenolic, flavonoid, and terpene compounds are abundant in *L. martabanica* roots, suggesting their role as active ingredients [17]. Based on this study, consistent compounds were found in the leaves of *L. martabanica*. Consequently, both the roots and leaves contain similar compounds, supporting the assumption that the active ingredients may be identical. Furthermore, it is possible that a cluster of terpenoids could serve as the active component in preventing pesticide toxicity [36]. Moreover, phenolic and flavonoid compounds possess antioxidant capabilities, which aid in the reduction of oxidative stress [37]. Consequently, it is plausible that these chemicals are the active components in *L. martabanica* leaf water extract that are responsible for the reduction of liver fibrosis activity.

In this study, the chemical composition of the water extract of *L. martabanica* leaf was analyzed using HPTLC with detection under UV light at 254 and 366 nm. An anisaldehyde-sulfuric acid, a universal reagent for natural products, and DPPH spraying reagent were also used. Evaluation using accepted chemical standards found the extract of *L. martabanica* to be free of caffeic acid, ellagic acid, gallic acid, kaempferol, quercetin, and rutin (Appendix A).

Chromatographic techniques can establish chemical profiles that are useful for quality control and determining the authenticity of raw materials and herbal products. These specifications can provide guidance for quality control and can serve as a reference for future research on the repeatability of the extract preparation process. Phytochemical screening revealed the presence of phenolics, flavonoids, saponins, and terpenoids. The bioactive compounds found in this plant will be used in future studies using bioassay-guided isolation.

Antioxidants are crucial for neutralizing free radicals and safeguarding cells against oxidative damage [38]. To comprehensively evaluate antioxidant capabilities and mechanisms, it is necessary to utilize three models, encompassing the DPPH and superoxide radical assays [39,40]. The outcomes from all antioxidant tests in this study demonstrated that the leaf water extract of *L. martabanica* displayed considerable antioxidant properties in both DPPH and superoxide radical assays. Moreover, previous research and literature reviews have highlighted the antioxidant activity present in plants of the genus *Litsea*, with this attribute being observed across different parts of the plants, including leaves, roots, and stems [41,42,43]. For example, the methanolic extract of *L. glutinosa* has demonstrated antioxidant properties, including the hydrogen peroxide scavenging activity, total antioxidant capacity, nitric oxide scavenging activity assay, and reducing power test [44]. The 80% ethanol extract and solvent fractions of *L. japonica* leaves have radical scavenging antioxidant activity [45]. Our own earlier experiments produced consistent results, confirming that the *L. martabanica* leaf water extract also exhibits antioxidant effects.

The liver is the major organ that deals with drugs and hazardous substances that can cause inflammation and oxidative stress [46]. Free radicals drive liver cell fibrosis, disrupting normal cellular processes, such as growth, division, and death regulation. This oxidative stress triggers Kupffer cells to release fibrosis-promoting factors, including TNF-α, IL-1β, IL-6, and TGF-β, while hepatic stellate cells produce collagen, leading to fibrosis [47]. Targeting hepatic stellate cell proliferation and promoting apoptosis offers potential in fibrosis treatment [20]. Compounds such as luteolin inhibit fibrosis-related genes, block cytokine pathways, and induce hepatic stellate cell apoptosis [48]. In our study, we utilized the LX-2 cell line, which possesses characteristics resembling hepatic stellate cells and is commonly used for investigating human hepatic fibrosis. Previous studies have shown that chlorpyrifos can induce liver fibrosis [18]. Mechanistically, after liver injury, hepatic stellate cells (HSCs) activate and transform into a myofibroblast-like state to repair the injury. These activated HSCs express α-smooth-muscle actin (α-SMA) and produce type-I collagen, a key component of the extracellular matrix (ECM). The mechanism of fibrosis caused by the pesticides mentioned above is similar to that in humans [49]. Therefore, this cell was chosen as a target for treating liver fibrosis [20]. The present study observed that the water extract, ethyl acetate, and butanol, as well as the residue of *L. martabanica* leaf at concentrations ranging from 200 to 800 µg/mL, significantly inhibited cell growth compared to control cells. The results also showed that *L. martabanica* leaf water extract tended to induce apoptosis, but only at very high concentrations. Ethyl acetate (800 µg/mL), butanol, and residue (400 and 800 µg/mL) were able to induce apoptosis cell death, demonstrating early apoptosis. The identification of the ethyl acetate fraction as an active ingredient presents an intriguing possibility for further investigation.

The objective of this test, however, was to verify the efficacy of the selected water extract. Its administration is intended to imitate traditional concoctions based on local knowledge, with the goal of expanding upon traditional knowledge and developing new products for future use. The present animal research study used conventional extraction methods, including water extraction, in order to conduct a more comprehensive examination of the impacts of *L. martabanica* leaf water extract.

The toxicity of organophosphate pesticides largely arises from inhibiting AChE, which results in an accumulation of acetylcholine [6]. *L. martabanica* root water extract has previously been shown to have anti-pesticide effects, such as improving AChE activity in rats exposed to chlorpyrifos [17]. In our study, the rats exposed to chlorpyrifos experienced a decrease in AChE activity. Treatment with *L. martabanica* leaf water extract demonstrated promise in reversing this effect, indicating potential as an anti-pesticide agent. Additionally, organophosphate pesticides are involved in oxidative stress and reactive oxygen species (ROS) production [50]. In our study, chlorpyrifos increased MDA levels while decreasing reduced glutathione, but treatment with *L. martabanica* leaf water extract reversed these effects. Furthermore, the *L. martabanica* leaf water extract was found to elevate SOD levels. According to prior studies, the methanol leaf extract of *L. glutinosa,* including neophytadiene (sesquiterpenoids), substantially increased SOD, GSH, and GPx concentrations while decreasing MDA, in comparison to the control group. Furthermore, it exhibited the potential to exert hepatoprotective effects via TGF-β1 signaling pathways [36]. Based on our study evaluating the quality of the extracts, antioxidant effects in vitro, and anti-pesticide properties in the animal study, it is plausible that the active ingredients belong to the terpenoids group.

Long-term exposure to high doses of organochloride and organophosphate pesticides commonly used in agriculture has been linked to persistent hematotoxicity and a higher incidence of aplastic anemia in humans [51,52]. Additionally, studies have demonstrated a significant decrease in hemoglobin concentration, RBC count, and hematocrit with increased exposure to chlorpyrifos [53]. The reduction in the red blood cell count could potentially be attributed to an elevated rate of erythrocyte destruction within the hematopoietic organ, erythropoiesis inhibition, hemosynthesis impairment, or osmoregulatory dysfunction [54]. In our investigation, we observed a decrease in RBCs, hematocrit, and hemoglobin levels across all chlorpyrifos-exposed groups. Despite some decreases in values, such as RBCs, hematocrit, and hemoglobin, after receiving various doses of the extract, they remained within the normal range [55,56,57,58]. Interestingly, a previous study reported that chlorpyrifos triggers a rise in white blood cells [53]. An increase in the total number of white blood cells (WBCs) caused by exposure to toxic substances, including pesticides, is a response to stress conditions [59]. Our investigation confirmed this relationship, showing an increase in white blood cells after chlorpyrifos exposure, with specific increases observed in neutrophils, lymphocytes, monocytes, and eosinophils, although without reaching statistical significance. Exposure to *L. martabanica* leaf water extract appeared to restore these white blood cell counts to normal levels, with all white blood cell parameters remaining within normal ranges [60].

To assess the potential impact on renal and hepatic functions, we conducted thorough clinical blood chemistry assessments. The kidneys, due to their high blood perfusion, are particularly susceptible to toxic substances. Such toxins can be actively filtered by the kidneys, potentially leading to their accumulation in renal tubules, emphasizing the importance of monitoring BUN and creatinine levels as sensitive indicators of renal health [61,62]. Most studies have reported that chlorpyrifos increases BUN and creatinine levels [63]. In our study, we observed a decrease in chlorpyrifos-induced BUN levels, while creatinine levels remained normal. This change does not imply abnormality from chlorpyrifos at this concentration. Despite some decreases in values, such as BUN, after administering the water extract at various doses, all values remained within normal ranges [57,58,61]. These findings suggest that the administration of *L. martabanica* leaf water extract had no discernible effect on kidney function or increase in kidney toxicity.

Chlorpyrifos leads to hepatotoxic changes by elevating AST, ALT, and ALP levels while reducing total protein and albumin [64,65]. Moreover, chlorpyrifos may cause liver dysfunction due to liver membrane permeability changes [66]. Our findings were in line with this, showing that chlorpyrifos increased AST levels but without statistical significance and significantly decreased total protein, with a trend toward decreasing albumin. Serum alkaline phosphatase activity (ALP) is a valuable indicator of liver disease, especially cholestatic disease. Additionally, ALP levels can serve as an indicator of bone mineral density loss [67]. Exposure to chlorpyrifos has been documented to affect chondrogenesis in the growth plate cartilage of long bones in chick embryos, disrupting ossification [68]. Our research showed a reduction in ALP levels following chlorpyrifos administration compared to the normal group. However, co-administration of *L. martabanica* leaf water extract with chlorpyrifos failed to normalize ALP levels. Thus, it is plausible that *L. martabanica* leaf water extract does not restore normal ossification or other bone processes. Furthermore, the extract does not induce toxicity in the liver or bones, as evidenced by the ALP value. However, all abnormal parameters of liver function tests remained within the normal range.

This experiment mimics the use of chemical pesticides by farmers in highland communities, who typically expose their plots continuously for 16 days to control pests. Therefore, pesticides were administered to the test animals for 16 days. However, in a previous study, the administration of pesticides over a period of around 28 days was shown to cause abnormality of hematological and biochemical parameters, leading to liver and kidney damage [66]. In our study, hence, it is possible that the duration of exposure was insufficient to induce abnormality in both hematological and biochemical parameters.

From the results of hematology and biochemical parameters, the values in all groups of animals were within the normal range. However, there were a few significantly different values of hematology and biochemical parameters between the control and treated groups. Thus, it was essential to conduct a histopathological examination to confirm and identify potential cellular-level abnormalities in these organs. Upon examination, the kidney cells appeared normal, while abnormalities were evident in the liver cells. In our study, rats administered chlorpyrifos (control group) exhibited abnormal morphology of cells, such as widened sinusoids and liver enlargement, compared to the normal group. In addition, the liver cells displayed hyperchromatic and hypertrophied nuclei with variations in size. Further, no dispersed cell necrosis or lymphocytic infiltration was observed. On the other hand, these pathology changes in liver cells gradually returned to the normal condition in the rats that received all concentrations of *L. martabanica* water extract, especially at high concentrations (750 and 250 mg/kg).

The administration of high concentrations of *L. martabanica* water extract led to the restoration of normal liver tissue characteristics. Regarding the pathology of the liver, *L. martabanica* potentially inhibits oxidative stress by facilitating the action of antioxidants, including MDA, which restore the liver to its normal state [69]. This is consistent with previous studies, which reported that neophytadiene, one type of sesquiterpenoids (terpenoids), was found to have protective effects [36]. Additionally, flavonoids and phenolic substances typically have antioxidant properties that aid in the reduction of liver fibrosis [70]. Our findings suggest that the water extract of *L. martabanica* may protect liver cells from oxidative damage, enhance AChE activity, and improve abnormalities in liver tissue in the chlorpyrifos-induced hepatotoxicity in rats.

## 5. Conclusions

The results of this investigation indicated that the leaves possess similar anti-pesticide potential to the roots, as intimated by traditional knowledge and wisdom of highland communities. Therefore, it can be inferred that *L. martabanica* leaf water extract has antioxidant activity, enhances AChE activity, and improves histopathological abnormalities in the liver. However, further investigation is required using chlorpyrifos and *L. martabanica* over longer periods. A safety evaluation of the potential of *L. martabanica* leaf water extract to mitigate the side effects of pesticides, e.g., acute and chronic toxicity tests, will also be needed. It is especially crucial to explore the active ingredients within the water extract of *L. martabanica* in depth to study the molecular mechanism. The use of *L. martabanica* leaf water extract could potentially play a role in promoting both environmental sustainability and human health by enhancing healthcare accessibility through its traditional uses to improve and promote the health quality of people after toxicity caused by chemical residues in food, potable water, and food crops.

## Figures and Tables

**Figure 1 foods-13-01695-f001:**
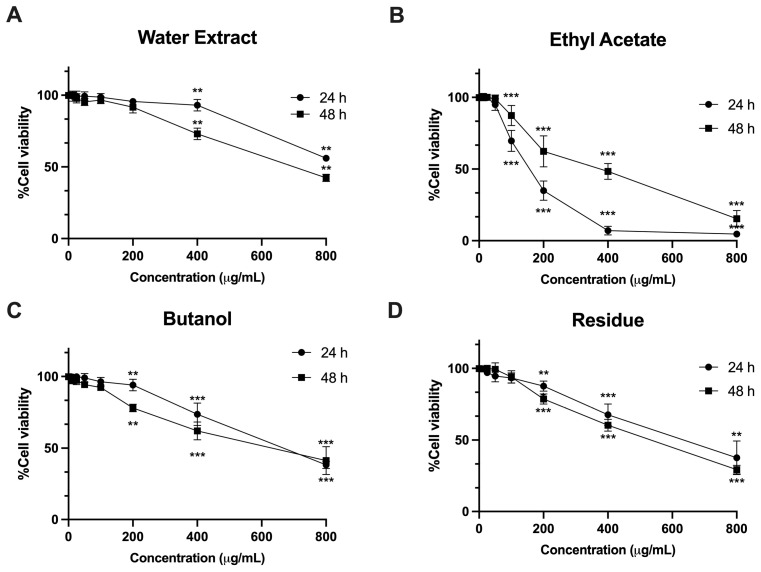
Cytotoxic effects of *L. martabanica* leaf water extract and its fractions on LX-2 cell lines. Cells were treated with various concentrations (12.5–800 µg/mL) of either *L. martabanica* leaf water extract (**A**) or its fractions ((**B**): ethyl acetate, (**C**): butanol, and (**D**): residue) for 24 and 48 h. Cell viability was assessed by comparison with 0.1% DMSO-treated control cells after 24 and 48 h of incubation. Results are presented as mean ± S.E.M. from three independent experiments. ** *p* < 0.01 and *** *p* < 0.001 vs. control.

**Figure 2 foods-13-01695-f002:**
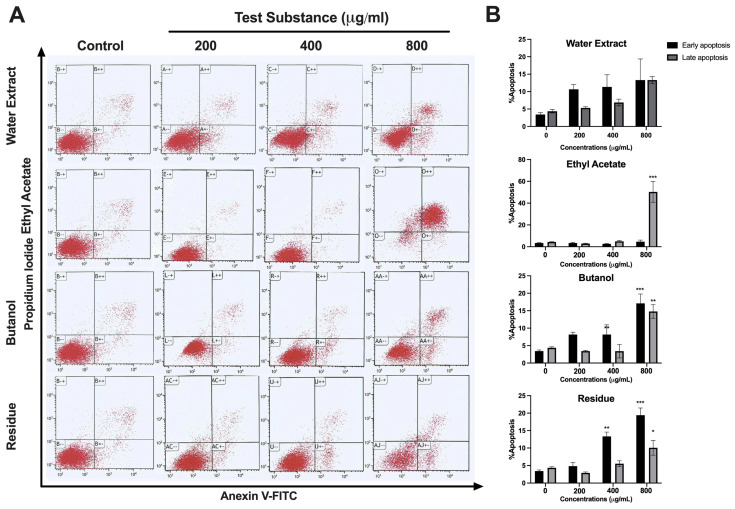
Effects of *L. martabanica* leaf water extract on apoptosis induction in LX-2 cell lines. Dot plot (**A**) and bar graph (**B**) show the flow cytometric analysis of apoptosis induction on LX-2 cells treated with various concentrations (200–800 µg/mL) of *L. martabanica* leaf water extract. Results are presented as mean ± S.E.M. from three independent experiments. * *p* < 0.05, ** *p* < 0.01, and *** *p* < 0.001 vs. control (one-way ANOVA with Tukey’s post hoc test).

**Figure 3 foods-13-01695-f003:**
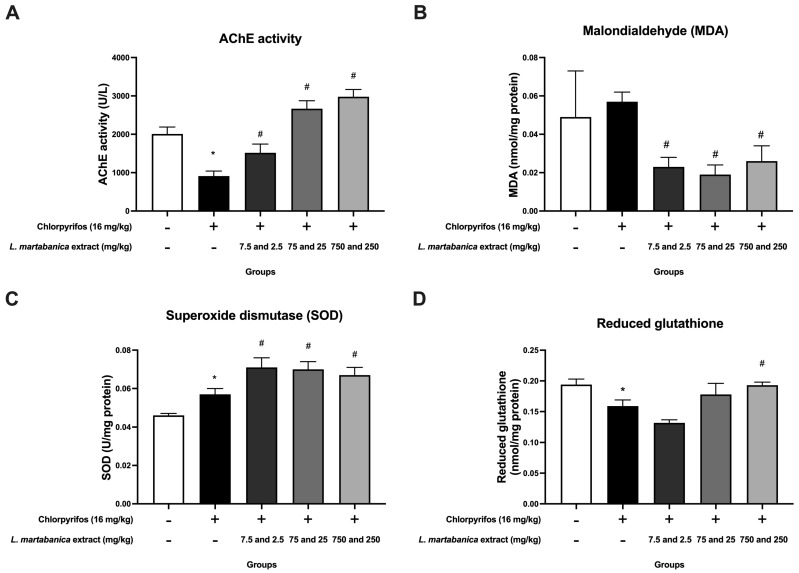
Effects of *L. martabanica* leaf water extract on anti-Pesticide potential in rats. AChE activity (**A**), malondialdehyde (**B**), superoxide dismutase (**C**), and reduced glutathione (**D**). Results are presented as mean ± S.E.M. * Statistically significantly different from the normal rats (distilled water), *p* < 0.05. # Statistically significantly different from the control group (distilled water + chlorpyrifos), *p* < 0.05.

**Figure 4 foods-13-01695-f004:**
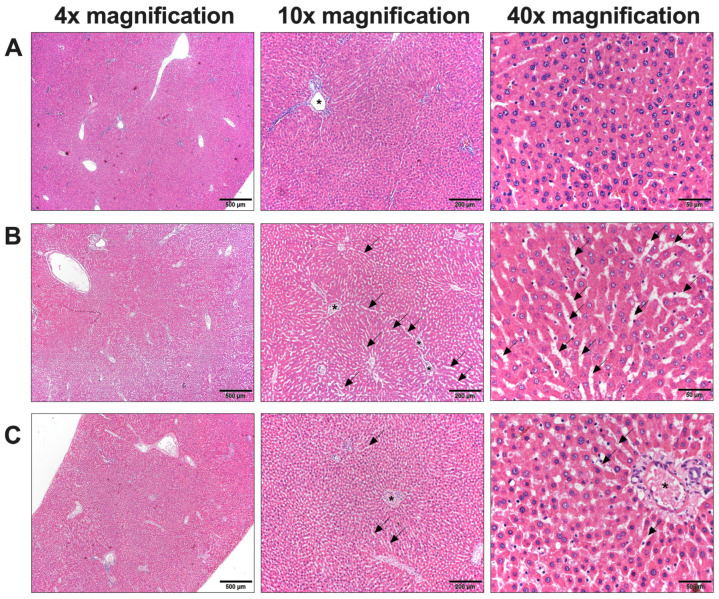
Histopathologic results of the rat liver. (**A**) Histology results of the normal rat liver in the normal group. (**B**) Histology results of the rat liver in the control group receiving chlorpyrifos. (**C**) Histology results of the rat liver in the group that received high doses of *L. martabanica* leaf water extract (750 and 250 mg/kg). The images indicate the hepatic portal vein (star) and sinusoidal dilatation (arrow).

**Figure 5 foods-13-01695-f005:**
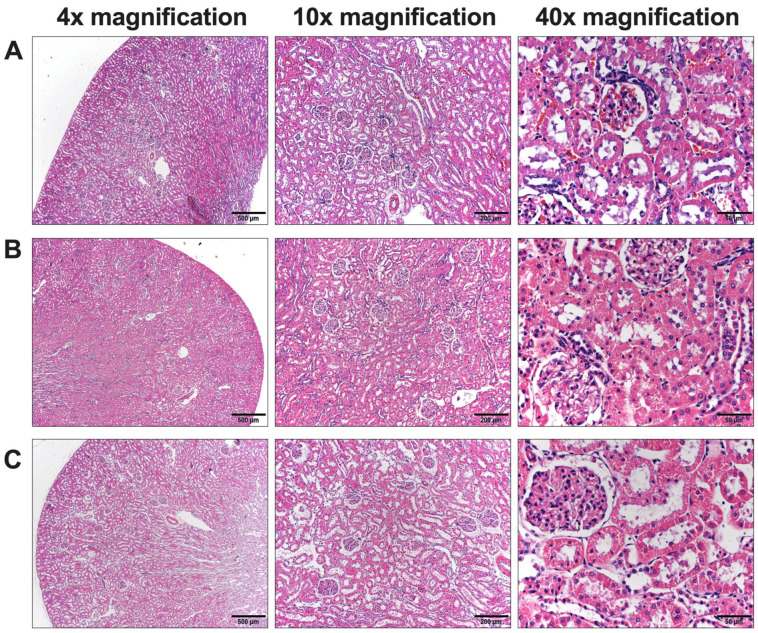
Histopathologic results of rat kidneys. (**A**) Histology results of rat kidneys in the normal group. (**B**) Histology results of kidneys in the control group receiving chlorpyrifos. (**C**) Histology results of kidneys in the group that received high doses of *L. martabanica* leaf water extract (750 and 250 mg/kg).

**Table 1 foods-13-01695-t001:** Pharmacognostic characteristics of *L. martabanica* leaves.

Specification	Content (%) by Dried Weight
Loss on drying	6.39 ± 0.04
Total ash	5.23 ± 0.01
Acid-insoluble ash	0.14 ± 0.01
Ethanol-soluble extractive value	9.57 ± 0.02
Water-soluble extractive value	13.19 ± 0.04

Values are expressed as mean ± S.E.M. from three independent experiments.

**Table 2 foods-13-01695-t002:** The IC_50_ values of *L. martabanica* leaf water extract in the DPPH and superoxide radical assays.

Sample	IC_50_ (μg/mL)	
	DPPH	Superoxide
*L. martabanica* leaf water extract	56.17± 3.5 ***	48.86 ± 4.9 *
Ethyl acetate fraction	35.3 ± 2.1 **	49.7 ± 3.0 *
Butanol fraction	45.7± 0.7 **	47.4 ± 2.4 *
Residue fraction	68.8± 10.9 ***	80.6 ± 9.7 ***
Gallic acid	2.70 ± 0.01	23.18 ± 3.9

Values are expressed as mean ± S.E.M. from three independent experiments. IC_50_, the half-maximal inhibitory concentration; DPPH, 2,20-diphenyl-1-picrylhydrazyl. * *p* < 0.05, ** *p* < 0.01, and *** *p* < 0.001 vs. gallic acid.

**Table 3 foods-13-01695-t003:** Determination of IC_50_ values of *L. martabanica* leaf water extract and its fractions.

Sample	IC_50_ (μg/mL)	
	24 h	48 h
*L. martabanica* leaf water extract	>1000	564.70 ± 40.95
Ethyl acetate fraction	154.7 ± 11.29	539.90 ± 38.26
Butanol fraction	572.90 ± 69.58	437.60 ± 22.57
Residue fraction	762.30 ± 10.74	690.00 ± 7.94

Values are expressed as mean ± S.E.M. from three independent experiments.

**Table 4 foods-13-01695-t004:** Effect of *L. martabanica* leaf water extract on body, liver, and kidney weights of rats.

Parameters	Normal	Control	*L. martabanica* Leaf Water Extract (mg/kg)		
			7.5 and 2.5	75 and 25	750 and 250
Body weights					
Day 1	242.50 ± 5.88	240.83 ± 3.27	243.33 ± 4.59	249.17 ± 4.73 *^,a^	250.00 ± 2.89 *^,a^
Day 8	270.00 ± 9.22	281.67 ± 4.22	270.00 ± 9.04	279.17 ± 6.88	279.17 ± 7.35
Day 16	263.33 ± 11.52	263.33 ± 8.53	275.00 ± 15.00	290.00 ± 8.16	295.00 ± 14.20
Organ weights					
Liver	11.60 ± 0.78	12.07 ± 0.31	14.32 ± 1.54	11.07 ± 0.46	10.83 ± 0.63
Kidney	1.34 ± 0.04	1.33 ± 0.06	1.62 ± 0.08 *^,a^	1.30 ± 0.04	1.45 ± 0.07

Values are expressed as mean ± S.E.M. from three independent experiments, *n* = 6. * Significantly different from the normal rats (distilled water), *p* < 0.05. ^a^ Significantly different from the control group (distilled water + chlorpyrifos), *p* < 0.05.

**Table 5 foods-13-01695-t005:** The hematological analysis of rats in the study of the anti-pesticide effect of *L. martabanica* leaf extract.

Parameters	Normal	Control	*L. martabanica* Leaf Water Extract (mg/kg)		
			7.5 and 2.5	75 and 25	750 and 250
Red blood cells (×10^6^/µL)	8.58 ± 0.22	8.20 ± 0.15	8.17 ± 0.23	7.59 ± 0.11 *	7.64 ± 0.08 *
Hemoglobin (g/dL)	15.42 ± 0.36	14.70 ± 0.20	14.85 ± 0.31	13.68 ± 0.19 *	13.57 ± 0.15 *^,a^
Hematocrit (%)	48.00 ± 1.23	45.33 ± 0.75	46.02 ± 0.92	42.32 ± 0.72 *	41.53 ± 0.29 *^,a^
Mean corpuscular volume (fL)	56.00 ± 0.93	55.38 ± 1.08	56.43 ± 0.70	55.78 ± 0.74	54.38 ± 0.38
Mean corpuscular hemoglobin (pg)	17.98 ± 0.35	17.95 ± 0.31	18.20 ± 0.20	18.05 ± 0.11	17.73 ± 0.23
Mean corpuscular hemoglobin concentration (g/dL)	32.15 ± 0.24	32.47 ± 0.18	32.27 ± 0.22	32.37 ± 0.27	32.65 ± 0.32
Platelet (×10^5^/µL)	8.26 ± 0.29	7.55 ± 0.29	8.12 ± 0.30	7.90 ± 0.27	9.59 ± 0.50 ^a^

Values are expressed as mean ± S.E.M. from three independent experiments, *n* = 6. * Significantly different from the normal rats (distilled water), *p* < 0.05. ^a^ Significantly different from the control group (distilled water + chlorpyrifos), *p* < 0.05.

**Table 6 foods-13-01695-t006:** White blood cell count analysis of rats in the study of the anti-pesticide effect of *L. martabanica* leaf extract.

Parameters	Normal	Control	*L. martabanica* Leaf Water Extract (mg/kg)		
			7.5 and 2.5	75 and 25	750 and 250
White blood cells (×10^3^ cells/µL)	6.24 ± 0.89	8.64 ± 0.49	7.45 ± 0.39	7.55 ± 0.56	7.88 ± 0.61
Neutrophil (cells/µL)	1.48 ± 0.25	1.78 ± 0.35	1.92 ± 0.16	1.79 ± 0.25	1.75 ± 0.36
Lymphocyte (cells/µL)	4.03 ± 0.75	5.72 ± 0.37	4.99 ± 0.31	5.13 ± 0.34	5.36 ± 0.41
Monocyte (cells/µL)	0.64 ± 0.12	0.96 ± 0.16	0.45 ± 0.05	0.54 ± 0.15	0.65 ± 0.15
Eosinophil (cells/µL)	0.10 ± 0.02	0.19 ± 0.03	0.09 ± 0.01	0.10 ± 0.04	0.13 ± 0.04
Basophil (cells/µL)	0.00 ± 0.00	0.00 ± 0.00	0.00 ± 0.00	0.00 ± 0.00	0.00 ± 0.00

Values are expressed as mean ± S.E.M. from three independent experiments, *n* = 6.

**Table 7 foods-13-01695-t007:** The blood chemical analysis of rats in the study of the anti-pesticide effect of *L. martabanica* leaf extract.

Parameters	Normal	Control	*L. martabanica* Leaf Water Extract (mg/kg)		
			7.5 and 2.5	75 and 25	750 and 250
BUN (mg/dL)	21.48 ± 3.03	17.00 ± 0.45	18.75 ± 0.67	14.48 ± 0.94	12.02 ± 1.08 *
Creatinine (mg/dL)	0.68 ± 0.04	0.62 ± 0.02	0.64 ± 0.03	0.64 ± 0.02	0.62 ± 0.01
Total protein (g/dL)	6.23 ± 0.18	5.62 ± 0.18 *	5.25 ± 0.08 *	5.10 ± 0.07 *	5.32 ± 0.08 *
Albumin (g/dL)	2.92 ± 0.06	2.80 ± 0.06	2.73 ± 0.03	2.80 ± 0.04	2.77 ± 0.03
Total bilirubin (mg/dL)	0.20 ± 0.02	0.19 ± 0.01	0.20 ± 0.01	0.19 ± 0.01	0.21 ± 0.00
Direct bilirubin (mg/dL)	0.06 ± 0.01	0.06 ± 0.00	0.06 ± 0.00	0.05 ± 0.00	0.06 ± 0.00
Aspartate aminotransferase (AST; U/L)	94.83 ± 5.31	102.83 ± 4.55	107.33 ± 13.04	78.33 ± 3.00	77.00 ± 3.07
Alanine aminotransferase (ALT; U/L)	33.00 ± 3.71	29.00 ± 1.21	30.50 ± 4.57	30.33 ± 2.35	25.67 ± 1.99
Alkaline phosphatase (U/L)	116.83 ± 6.95	83.17 ± 7.66 *	85.67 ± 11.13	76.50 ± 7.33	70.17 ± 6.54 *

Values are expressed as mean ± S.E.M. from three independent experiments, *n* = 6. * Significantly different from the normal rats (distilled water), *p* < 0.05.

## Data Availability

The original contributions presented in the study are included in the article/Appendix A, further inquiries can be directed to the corresponding author.

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
