# Peer review of "The Ameliorative Effect of Litsea martabanica (Kurz) Hook. f. Leaf Water Extract on Chlorpyrifos-Induced Toxicity in Rats and Its Antioxidant Potentials"

_foods, 2024, doi:10.3390/foods13111695_

Round 1

Reviewer 1 Report

Comments and Suggestions for Authors

Minor revision

This study aims to evaluate the antioxidant activity and pesticide resistance potential of  L. martabanica leaves instead of its roots through in vitro and in vivo experiments. In general, the design of the paper is reasonable and logical, but the paper still has many shortcomings, and writing is not easy to understand.

Major comments:

1. The authors should specify the plant's growth year and season of harvesting in Section 2.4.

2. It is recommended to supplement thin-layer chromatography profiles.

3. The unit of time should be labeled in Figure 1.

4. The selection of hepatic stellate cells should be explicitly stated in the article.

5. It is suggested to add "*p < 0.05, and **p < 0.01" in Figure 2.

6. Arrow annotations are recommended for indicating the location of histopathological lesions in Figure 5.

Author Response

Minor revision

This study aims to evaluate the antioxidant activity and pesticide resistance potential of L. martabanica leaves instead of its roots through in vitro and in vivo experiments. In general, the design of the paper is reasonable and logical, but the paper still has many shortcomings, and writing is not easy to understand.

Response: We are grateful to you for your recommendation. We fully revised the introduction part to make the material simpler and more understandable.

Major comments:

  1. The authors should specify the plant's growth year and season of harvesting in Section 2.4.

Response: We appreciate your guidance. The year and season of harvesting have been added in Section 2.2. (Line 100)

  1. It is recommended to supplement thin-layer chromatography profiles.

Response: We sincerely appreciate your comment. TLC Chromatogram was added to the supplement.

  1. The unit of time should be labeled in Figure 1.

Response: Thank you very much for your remark. The hours-based units of time have been added to Figure 1.

  1. The selection of hepatic stellate cells should be explicitly stated in the article.

Response: We have indicated and shown the reason why we selected this cell in the discussion sector: “In our study, we utilized the LX-2 cell line, which possesses characteristics resembling hepatic stellate cells and is commonly used for investigating human hepatic fibrosis. Previous studies have shown that chlorpyrifos can induce liver fibrosis (1). Mechanistically, after liver injury, hepatic stellate cells (HSCs) activate and transform into a myofibroblast-like state to repair the injury. These activated HSCs express α-smooth muscle actin (α-SMA) and produce type-I collagen, a key component of the extracellular matrix (ECM). The mechanism of fibrosis caused by the pesticides mentioned above is similar to that in humans (2). Therefore, the research team chose to use these cells as a target for treating liver fibrosis.The present study observed that water extracts, ethyl acetate, butanol, and the residue of L. martabanica leaf at concentrations ranging from 200 to 800 µg/mL significantly inhibited cell growth compared to control cells.” (Lines 495-503)

Reference:

(1) Abbasi, Aftab, Samreen Memon, and Salman Ahmed Farsi Kazi. “antifibrotic effects of angiotensin receptor blockers in organophosphate (chlorpyrifos) induced liver fibrosis.” Journal of Population Therapeutics and Clinical Pharmacology 31.1 (2024): 1302-1315.

(2) Acharya, P.; Chouhan, K.; Weiskirchen, S.; Weiskirchen, R. Cellular Mechanisms of Liver Fibrosis. Frontiers in Pharmacology 2021, 12, doi:10.3389/fphar.2021.671640.

  1. It is suggested to add "*p < 0.05, and **p < 0.01" in Figure 2.

Response: Thank you for your pointing out. The p-values have been annotated as follows: * p < 0.05; ** p < 0.01; *** p < 0.001 in comparison to the control one-way ANOVA using Tukey's post hoc test.

  1. Arrow annotations are recommended for indicating the location of histopathological lesions in Figure 5.

Response: Thank you for pointing that out. According to the results of the histopathological examination, the kidneys of the rats did not exhibit any abnormalities; hence, no markers were included in Figure 5. However, histopathology analysis is presented due to the organ's critical role in pesticide detoxification. In addition, we have added the histopathology description of the kidneys in the results section.

Reviewer 2 Report

Comments and Suggestions for Authors

This study seeks to assess the antioxidant activity and anti-pesticide potential of water extracts from L. martabanica leaves as an alternative to its roots, employing both in vitro and in vivo methodologies. Results revealed varied antioxidant activity across L. martabanica water extracts and fractions, with the water extract obtained via traditional methods selected for in vivo experimentation. Rats administered this extract prior to chlorpyrifos exposure exhibited diminished acetylcholinesterase activity compared to chlorpyrifos-exposed rats without treatment. Conversely, treatment with leaf extract resulted in elevated AChE activity, suggesting its efficacy as a detoxifying agent. Histopathological examination indicated the absence of liver cell necrosis and evidenced liver cell regeneration, implying potential protection against free radical-induced liver damage without adverse effects on liver weight or biochemical profiles. Utilizing L. martabanica leaves for medicinal purposes presents a promising avenue without exacerbating environmental concerns associated with root harvesting.

The article is well-structured, featuring an adequate number of experiments that are meticulously designed and accompanied by sound statistical analysis. Nevertheless, several experiments presented lack detailed descriptions in the methodology section, with some being superficially addressed. Moreover, the discussion section lacks depth and fails to thoroughly analyze the results. Furthermore, there appears to be some inconsistency between the histological and hematological findings and the discussion, as the authors did not observe any hepatic, renal, or hematological damage with this model. Therefore, I suggest a thorough review of the discussion section. Additionally, there are several other details that I have listed:

1.   Line 42: The authors assert widespread global utilization, yet rely on a citation from 2009. To bolster their claim, it is imperative for them to update and supplement their references with more recent evidence.

2.   Line 49: include a current quotation to give more strength to the statement

3. Line 66: Why are only these two compounds referenced? Are no other reactive oxygen and nitrogen species generated? Or is this specificity exclusive to this organophosphate?.

4.  Line 69-71: This concept is closely tied to and repetitious of the earlier statement in the paragraph. It would be appropriate to rephrase it and be more concise.

5.    Line 125: It is not in italics as in the previous subtitles.

6.    Line 205: Concentrations should be indicated.

7.   Line 219: It is essential to indicate the concentrations tested

8.  Line 296: Why is this not described briefly while MDA and the SOD method are detailed?

9. Line 320 (Table 1): What is the rationale for listing the content of compounds in this table without indicating the percentage? The table refers to quantitative values, not qualitative ones.

10. Line 331-332: As the concentration of the extract is not specified, there is no basis for comparison among the various fractions. It is essential to provide this information, especially considering that the methodology does not mention the utilization of different fractions with distinct solvents.

11.   Line 367: does not indicate the value of p?

12.   Line 377: It is not described in the methodology how it was performed.

13.   Line 388: It is not described in the methodology how it was performed.

14.   Line 391: This represents a range of concentrations. Therefore, elucidating how to interpret the graphed values is necessary. Are they depicted as an average within this range? If so, the range is too extensive to have such small standard deviations.

15.   Line 397-399: This is a conclusion, it should not be included in the results.

16.   Line 401: Rephrase the statement as it currently suggests that the figure title solely pertains to AChE activity.

17.   Line 406-413: Why does it not reference liver values and only mention the kidney?

18. Line 414: It would be the positive control? because it already has a control with the normal group. Same observation, if it is the average of the ranges, how can you have such small deviations.

19.   Line 418: There is no indication in the methodology as to how it has been carried out.

20.   Line 420:  75 mg, it is already a different value than the one mentioned above.

21.   Line 437: The methodology did not describe how it was carried out.

22.   Line 451: The method only mentions that they are left in formalin, without any further description.

23.   Line 458: Only what was observed at these doses is reported, and what was observed at the other doses?

24.   Line 459: The description is very vague, no reference is made to what was observed at different concentrations.

25.   Line 508-509: It would be advantageous for them to reference what was found in the root concerning these compounds—whether they are similar, different, or in what proportions. This aspect requires further discussion.

26.   Line 595-596: Although the values fall within the normal range, there was a statistically significant decrease in BUN concentration from 750 to 250, compared to the control, suggesting hepatic damage. However, transaminases and albumin levels remain unchanged. Furthermore, there are no observed changes in the positive control treated with chlorpyrifos, where the BUN value is not as low. What could account for this discrepancy?

27.   Line 598: This conclusion may not hold true as the positive control seemingly did not induce hepatic or renal damage either, thereby lacking a suitable positive control. What explanation could be provided for this?

28.   Line 601: An appropriate explanation must be provided to account for the lack of an effect, as the trend alone does not justify damage.

29.   Line 617: How do you justify that the value is within the normal range when it is below the positive control value? Once again, it appears that with this subchronic dosing model, hepatic damage is not sufficiently induced.

30.   Line 621-622: This assertion should have been made earlier, as it is evident that hepatic and renal damage is not present. Merely stating that values fall within the normal range without further discussion is insufficient justification.

31.   Line 624: This assertion contradicts the information presented in the preceding paragraph and the hematological, hepatic, and renal results.

32.   Line 631: It is not appropriate to discuss a regenerative process, as the observed tissue damage is minimal and consistent with normal levels of transaminases measured in blood.

33.   Line 631: This parameter was quantified in blood, not in hepatic tissue, thus it would be speculative to correlate it directly with the liver.

34.   Line 641-642: Again, it is speculative to draw this conclusion, as the model utilized does not demonstrate a well-established hematological, renal, and hepatic damage with the dosage and duration employed.

35.   Line 646: This cannot be concluded from the results shown.

Author Response

This study seeks to assess the antioxidant activity and anti-pesticide potential of water extracts from L. martabanica leaves as an alternative to its roots, employing both in vitro and in vivo methodologies. Results revealed varied antioxidant activity across L. martabanica water extracts and fractions, with the water extract obtained via traditional methods selected for in vivo experimentation. Rats administered this extract prior to chlorpyrifos exposure exhibited diminished acetylcholinesterase activity compared to chlorpyrifos-exposed rats without treatment. Conversely, treatment with leaf extract resulted in elevated AChE activity, suggesting its efficacy as a detoxifying agent. Histopathological examination indicated the absence of liver cell necrosis and evidenced liver cell regeneration, implying potential protection against free radical-induced liver damage without adverse effects on liver weight or biochemical profiles. Utilizing L. martabanica leaves for medicinal purposes presents a promising avenue without exacerbating environmental concerns associated with root harvesting.

The article is well-structured, featuring an adequate number of experiments that are meticulously designed and accompanied by sound statistical analysis. Nevertheless, several experiments presented lack detailed descriptions in the methodology section, with some being superficially addressed. Moreover, the discussion section lacks depth and fails to thoroughly analyze the results. Furthermore, there appears to be some inconsistency between the histological and hematological findings and the discussion, as the authors did not observe any hepatic, renal, or hematological damage with this model. Therefore, I suggest a thorough review of the discussion section. Additionally, there are several other details that I have listed:

  1. Line 42: The authors assert widespread global utilization yet rely on a citation from 2009. To bolster their claim, it is imperative for them to update and supplement their references with more recent evidence.

Response: We are grateful for the guidance. Through the addition of the following references, we have kept the information section on pesticide use up to date so that it more accurately reflects global use.

The updated reference: Riedo, J.; Wächter, D.; Gubler, A.; Wettstein, F.E.; Meuli, R.G.; Bucheli, T.D. Pesticide residues in agricultural soils in light of their on-farm application history. Environ Pollut 2023, 331, 121892, doi: 10.1016/j.envpol.2023.121892. (ref.1)

  1. Line 49: include a current quotation to give more strength to the statement

Response: We are grateful for the guidance. We have added a current quotation to support this statement.

Reference:

Neurotoxic effects: Vellingiri B., et al. (2022). Neurotoxicity of pesticides – A link to neurodegeneration. Ecotoxicol Environ Saf. 243:113972. doi: 10.1016/j.ecoenv.2022.113972. (ref. 2).

Carcinogenic effects: Pedroso T.M.A., et al. (2022). Cancer and occupational exposure to pesticides: a bibliometric study of the past 10 years. Environ Sci Pollut Res Int. 29(12):17464-17475. doi: 10.1007/s11356-021-17031-2. (ref. 3)

Heppatotoxic effects: Manfo F.P.T., et al. (2020). Evaluation of the Effects of Agro Pesticides Use on Liver and Kidney Function in Farmers from Buea, Cameroon. J Toxicol. 2305764. doi: 10.1155/2020/2305764. eCollection 2020. (ref. 4)

Immunotoxic effects: Lee, G.-H., & Choi, K.-C. (2020). Adverse effects of pesticides on the functions of immune system. Comparative Biochemistry and Physiology Part C: Toxicology & Pharmacology, 108789. doi:10.1016/j.cbpc.2020.10878910.1016/j.cbpc.2020.108789 (ref. 5).

  1. Line 66: Why are only these two compounds referenced? Are no other reactive oxygen and nitrogen species generated? Or is this specificity exclusive to this organophosphate?

Response: Thank you for your pointing out. In fact, the execution of organophosphate results in the production of reactive species, such as hydrogen peroxide (H2O2), superoxide anion (O2•−), reactive nitrogen species (RNS), and lipid peroxidation products (1). However, the primary source of organophosphate exposure is reactive oxygen species (ROS), hence in this work, we aimed to investigate the effects of L. martabanica water extracts on this pathway. For the benefit of the information's rationality, we therefore include RNS in the introduction part. (Lines 50-51)

Reference: (1) Abd El-Moneim Ibrahim, Khairy, et al. "Single or combined exposure to chlorpyrifos and cypermethrin provoke oxidative stress and downregulation in monoamine oxidase and acetylcholinesterase gene expression of the rat’s brain." Environmental Science and Pollution Research 27 (2020): 12692-12703. (ref. 11)

  1. Line 69-71: This concept is closely tied to and repetitious of the earlier statement in the paragraph. It would be appropriate to rephrase it and be more concise.

Response: We have fully revised and rewritten this statement in the introduction.

  1. Line 125: It is not in italics as in the previous subtitles.

Response: Thank you for your pointing out. From the template, the subsection was in italics, while the subsubsection was not in italics. We have already corrected it according to the template.

  1. Line 205: Concentrations should be indicated.

Response: We have included all concentrations of L. martabanica water extracts (25 – 200 µg/mL). (Line 137)

  1. Line 219: It is essential to indicate the concentrations tested.

Response: We have included all concentrations of L. martabanica water extracts (25 – 200 µg/mL). (Line 152)

  1. Line 296: Why is this not described briefly while MDA and the SOD method are detailed?

Response: We apologize for the insufficient details and missing information about the methods used to measure MDA, SOD, and reduced glutathione. To provide more information about the kits used for these assays, we have reorganized this section to include the testing methods based on these kits. (Line 230)

  1. Line 320 (Table 1): What is the rationale for listing the content of compounds in this table without indicating the percentage? The table refers to quantitative values, not qualitative ones.

Response: Thank you very much for your comment. We have already corrected the data in Table 1. It is merely a qualitative report on phytochemical group data from phytochemical screening. (Line 269)

  1. Line 331-332: As the concentration of the extract is not specified, there is no basis for comparison among the various fractions. It is essential to provide this information, especially considering that the methodology does not mention the utilization of different fractions with distinct solvents.

Response: In this research, we aimed to use water extracts to test the effects and find scientific information to support the traditional knowledge of highland communities. However, since the research team recognized the potential of this plant to be developed into a health product in the future, the extract was separated and fractioned, and these fractions were tested to compare their activity with the water extract as a guideline for identifying bioactive compounds in the future.

  1. Line 367: does not indicate the value of p?

Response: The text "*p < 0.05, **p < 0.01, and ***p < 0.001 vs control." has been added. (Lines 320-321)

  1. Line 377: It is not described in the methodology how it was performed.

Response: We apologize for the insufficient details and missing information about the methods used to measure MDA, SOD, and reduced glutathione. To provide more information about the kits used for these assays, we have reorganized this section to include the testing methods based on these kits. (Line 230)

These sentences have been added to the method section: “The serum was then analyzed for malondialdehyde (MDA) levels using an enzyme-linked immunosorbent assay (ELISA) kit from Elabscience (Houston, TX, USA), for superoxide dismutase (SOD) levels using the RANSOD kit (Randox Lab, Crumlin, UK), and for reduced glutathione levels using a Sigma Aldrich Glutathione Assay Kit (Millipore Sigma, Burlington, MA, USA; cat. no. MAK364), following the manufacturer’s protocol.”

  1. Line 388: It is not described in the methodology how it was performed.

Response: The procedure details for reducing glutathione levels are provided in the method section. (Line 230)

  1. Line 391: This represents a range of concentrations. Therefore, elucidating how to interpret the graphed values is necessary. Are they depicted as an average within this range? If so, the range is too extensive to have such small standard deviations.

Response: If the reviewer's question refers to the dose of the extract at concentrations of 75 and 25 mg/kg. This information represented the concentration of extracts. The administration of the leaf extract acted as an imitation of the concoction based on traditional knowledge used in highland communities. In group 3 (low dose), rats were subjected to a cyclic dosing regimen of L. martabanica leaf water extract, receiving a cyclical daily administration of 7.5 mg/kg for 2 days followed by 2.5 mg/kg for 2 days over a period of 16 days (four rounds). Group 4 (middle dose) received a cyclic dose of 75 mg/kg for 2 days followed by 25 mg/kg for 2 days, and group 5 (high dose) received a cyclic dose of 750 mg/kg for 2 days followed by 250 mg/kg for 2 days, also with daily administration over 16 days (four rounds). The dose for all three groups was not calculated for statistical comparison between groups. However, comparisons between groups were made using the results of MDA, SOD, and reduced glutathione, which are presented as mean ± S.E.M. These values can be displayed in graphical form using the SPSS program, and small standard deviations are calculated accordingly. In addition, we additionally address how each test group was administered to animals in the methods section.

  1. Line 397-399: This is a conclusion; it should not be included in the results.

Response: Thank you for your recommendation. We have deleted these sentences.

  1. Line 401: Rephrase the statement as it currently suggests that the figure title solely pertains to AChE activity.

Response: The title of this figure has been revised. (Line 351)

  1. Line 406-413: Why does it not reference liver values and only mention the kidney?

Response: The results obtained from the leaf water extract of L. martabanica revealed no abnormalities. There were no statistically significant alterations in the weight of the liver. As a consequence, the sentence has been added to the results section. (Lines 362-363)

  1. Line 414: It would be the positive control? because it already has a control with the normal group. Same observation, if it is the average of the ranges, how can you have such small deviations.

Response: The normal group (Normal) was orally administered 2 mL/kg of distilled water daily for 16 days. The control group (Control) received the same dose of distilled water plus chlorpyrifos at a dosage of 16 mg/kg daily for the same 16-day period. This experiment will compare the effects of the extracts on reducing markers related to the liver fibrosis process and improving detoxification. In addition, there is currently no standard medicine for treating liver fibrosis. Thus, this research report does not include a positive control group treated with such medicine. Therefore, the term "Control group" should be used in this article. In addition, all groups of rats received daily dosages rather than doses within range. This information is provided in the method section and the question 14. Furthermore, the exposures for all extracts in each group of rats were not in a range but rather a set of specific exposures, as described in the methods section. In addition, statistical data are presented as mean ± S.E.M. (standard error of the mean). The S.E.M. value is small because the raw data results are similar, resulting in small deviations.

  1. Line 418: There is no indication in the methodology as to how it has been carried out.

Response: For hematology analysis, whole blood was collected in ethylenediaminetetraacetic acid (EDTA) containing tube and measured using an automated hematology analyzer (BC-5300 Vet, Mindray, Shenzhen, China) provided by the Small Animal Hospital at Chiang Mai University. This information will be included in the method section. (Line 237)

  1. Line 420:  75 mg, it is already a different value than the one mentioned above.

Response: In this study, low dose concentrations of 7.5 and 2.5 mg/kg, middle doses of 75 and 25 mg/kg, and high doses of 750 and 250 mg/kg were used. We have checked the consistency of all contents to ensure the correctness of the information.

  1. Line 437: The methodology did not describe how it was carried out.

Response: Blood was collected in a clot-activated tube for biochemical analysis. Following centrifugation at 3,500 rpm for 10 minutes, the blood biochemical parameters were determined using an automated BX-3010 analyzer (Sysmex, Kobe, Japan) provided by the Small Animal Hospital at Chiang Mai University. This information will be included in the method section. (Line 237)

  1. Line 451: The method only mentions that they are left in formalin, without any further description.

Response: We have apologized for our missing information. After the organ was fixed in 10% formalin, fine needle aspirates were obtained, and the tissues were embedded in paraffin and stained with H&E. We consequently added a sentence describing the H&E staining procedure to the Methods section. Then, using a microscope, examine the morphology and numerous changes in the cells. (Line 245)

  1. Line 458: Only what was observed at these doses is reported, and what was observed at the other doses?

Response: For kidneys, rats administered pesticides in combination with different concentrations of L. martabanica leaf water extract did not exhibit any discernible morphological or histopathological abnormalities in their kidneys. Meanwhile, in the liver histopathology analysis, all concentrations of L. martabanica water extract could protect liver cells from chlorpyrifos-indued hepatotoxicity, especially at high concentrations (750 and 250 mg/kg) (Lines 411-413). In addition, this study demonstrates that high concentrations of these extracts do not induce hepatotoxicity while simultaneously restoring the liver to its normal state. We have modified the results of the histopathology analysis.

  1. Line 459: The description is very vague, no reference is made to what was observed at different concentrations.

Response: We have included the description of the histopathology results for the kidney. “Leucocyte infiltration, glomerulus atrophy, and renal tubule vacuolization were not observed in rats that were exposed to chlorpyrifos. Additionally, rats that received all concentrations of the L. martabanica water extracts along with chlorpyrifos did not show any abnormality in the kidney.” (Lines 414-418)

  1. Line 508-509: It would be advantageous for them to reference what was found in the root concerning these compounds—whether they are similar, different, or in what proportions. This aspect requires further discussion.

Response: We have added details from a previous study regarding the substances found in the roots to the discussion section. Both the roots and leaves of Litsea martabanica were found to contain phenolics, flavonoids, and terpenes. Additionally, apigenin, caffeic acid, and gallic acid were not found. Our study is consistent with the previous findings. (Lines 449-453)

  1. Line 595-596: Although the values fall within the normal range, there was a statistically significant decrease in BUN concentration from 750 to 250, compared to the control, suggesting hepatic damage. However, transaminases and albumin levels remain unchanged. Furthermore, there are no observed changes in the positive control treated with chlorpyrifos, where the BUN value is not as low. What could account for this discrepancy?

Response: Thank you for your pointing out. It is the kidney function that is represented by BUN, not the liver function. This experiment mimics the use of chemical pesticides by farmers in highland communities, who typically expose their plots continuously for 16 days to control pests. Therefore, to imitate the practice of farmers on farms, this study administered pesticides to the test animals for 16 days. Our results showed that even some hematological and biochemical blood parameters were significantly different from those in the control group. However, none of the parameters indicate abnormalities in the organs of the experimental animals because all values are within the normal range. If the values change but remain within the normal range, it is not considered an organ abnormality. Generally, for an abnormality to be clinically significant, the value must more than one-fold increase or decrease. However, in our study, the histopathological examination showed that the kidney was normal, but the liver was abnormal. Therefore, it is expected chlorpyrifos could stimulate abnormality at the histopathology level without any significant change in hematological or biochemical parameters.

In a previous study, the administration of pesticides over a period of around 28 days has been demonstrated to cause liver and kidney damage. For example, the study administered chlorpyrifos at a dose of 5 mg/kg over 28 days, observing changes in AST, ALT, albumin, and total protein (1). Therefore, research has indicated that the principal determinants of liver and kidney toxicity induced by pesticides are probably the duration and concentration of exposure. In our study, the high dose of chlorpyrifos was 16 mg/kg, but the duration was only 16 days. Therefore, it is possible that the duration of exposure was insufficient to induce toxicity. This section was incorporated into the discussion section. (Lines 581-593)

Reference:

(1) Uzun, Fatma Gokce, and Yusuf Kalender. "Chlorpyrifos induced hepatotoxic and hematologic changes in rats: the role of quercetin and catechin." Food and chemical toxicology 55 (2013): 549-556. (ref. 66)

  1. Line 598: This conclusion may not hold true as the positive control seemingly did not induce hepatic or renal damage either, thereby lacking a suitable positive control. What explanation could be provided for this?

Response: Experimental animals received chlorpyrifos and there were negligible changes in hematology or biochemical parameters. However, chlorpyrifos can stimulate abnormalities in the liver in histopathological analysis. These abnormalities include dilation or widening of the sinusoids, liver hypertrophy, and cells of varying shapes and sizes with hyperchromatic and hypertrophied nuclei without necrosis. In addition, it was found that all doses of the extract were able to return the liver to normal, with the most significant improvements observed at the highest dose (Figure 4).Therefore, L. martabanica water extracts have the potential to improve the abnormality of histopathology in the liver. (Lines 589-593)

  1. Line 601: An appropriate explanation must be provided to account for the lack of an effect, as the trend alone does not justify damage.

Response: I agree and thank you for the advice. These explanations have been discussed more in Lines 581-593.

  1. Line 617: How do you justify that the value is within the normal range when it is below the positive control value? Once again, it appears that with this subchronic dosing model, hepatic damage is not sufficiently induced.

Response: The animal laboratory must have a range of standard values in both hematological and biochemical parameters for animals of all breeds, genders, and ages. This study compared a group of chlorpyrifos-treated rats (control group) to all treated groups. If the results show a statistically significant increase or decrease in these parameterscompared to control group but remain within the range of standard values. It can be justified that these parameters are within the normal range. Even if hematological and biochemical parameters are significantly different and within the normal range, it is still important to confirm whether the organs related to these parameters are normal or not. Consequently, gross and histopathological analyses are essential parts of this consideration process, providing vital information for the comprehensive assessment of a pathological condition.

  1. Line 621-622: This assertion should have been made earlier, as it is evident that hepatic and renal damage is not present. Merely stating that values fall within the normal range without further discussion is insufficient justification.

Response: This experiment mimics the use of chemical pesticides by farmers in highland communities, who typically expose their plots continuously for 16 days to control pests. Therefore, pesticides were administered to the test animals for 16 days. In this study, experimental animals received chlorpyrifos, and although there were negligible changes in hematological or biochemical parameters, chlorpyrifos can stimulate abnormalities in the liver by histopathological analysis. These abnormalities include dilation or widening of the sinusoids, liver hypertrophy, and cells of varying shapes and sizes with hyperchromatic and hypertrophied nuclei without necrosis. In addition, it was found that all doses of the extract were able to return the liver to normal, with the most significant improvements observed at the highest dose.

Generally, in other investigations, the administration of pesticides over a period of around 28 days has been demonstrated to cause liver and kidney damage by biochemical parameters and histopathology analysis (1). Therefore, research has indicated that the principal determinants of liver and kidney toxicity induced by pesticides are probably the duration and concentration of exposure. These sections have been included in the discussion section.

Reference: (1) Uzun, Fatma Gokce, and Yusuf Kalender. "Chlorpyrifos induced hepatotoxic and hematologic changes in rats: the role of quercetin and catechin." Food and chemical toxicology 55 (2013): 549-556. (ref. 66)

  1. Line 624: This assertion contradicts the information presented in the preceding paragraph and the hematological, hepatic, and renal results.

Response: The reviewer is correct in stating that what the research team examines does not show any abnormalities of the organ because the hematological or biochemical parameters do not increase or decrease abnormally. However, statistical comparisons assume that there have been significant differences between the control group and other treated groups. As a result, it is important to clarify and identify the information on whether the differences occur as a result of organ abnormalities or not. Additionally, to gather information about the potential of medicinal plants, gross and histopathological examinations of the liver and kidneys were conducted, as these are key organs involved in the metabolism and excretion of toxins.

Therefore, we have modified and discussed this paragraph as follows: “From the results of hematology and biochemical parameters, the values in all groups of animals were within the normal range. However, there were a few significantly different values of hematology and biochemical parameters between the control and treated groups. Therefore, it was essential to conduct a histopathological examination to confirm and identify potential cellular-level abnormalities in these organs.” (Lines 589-593)

  1. Line 631: It is not appropriate to discuss a regenerative process, as the observed tissue damage is minimal and consistent with normal levels of transaminases measured in blood.

Response: I agree and thank you for the advice. The phrase "regenerative process" has been deleted.

  1. Line 631: This parameter was quantified in blood, not in hepatic tissue, thus it would be speculative to correlate it directly with the liver.

Response: As indicated in the previous discussion, these parameters assess histopathology in liver tissue through staining to identify normal and abnormal characteristics; this does not involve a blood test.

  1. Line 641-642: Again, it is speculative to draw this conclusion, as the model utilized does not demonstrate a well-established hematological, renal, and hepatic damage with the dosage and duration employed.

Response: This experiment mimics the use of chemical pesticides by farmers in highland communities, who typically expose their plots continuously for 16 days to control pests. Therefore, pesticides were administered to the test animals for 16 days. In this study, experimental animals received chlorpyrifos, and although there were no obvious changes in hematological or biochemical parameters, chlorpyrifos can stimulate abnormalities in the liver by histopathological analysis. These abnormalities include dilation or widening of the sinusoids, liver hypertrophy, and cells of varying shapes and sizes with hyperchromatic and hypertrophied nuclei without necrosis. In addition, it was found that all doses of the extract were able to return the liver to normal, with the most significant improvements observed at the highest dose. (Lines 589-593)

  1. Line 646: This cannot be concluded from the results shown.

Response: As the reviewer’s recommendation, the summary will be revised so that it is more suitable for the situation. (Lines 609-611)

Reviewer 3 Report

Comments and Suggestions for Authors

This study aimed to investigate the amelioration effect of Litsea martabanica leaf water extract on chlorpyrifos-induced toxicity. The purpose of this study is interesting, and this is a topic of interest to readers, but this study fails to explain the molecular mechanisms involved. Therefore, I recommend that this manuscript undergoes a major revision before judging its suitability for publication in the journal Foods.

(1) The authors need to deepen their research on the molecular mechanisms by which Litsea martabanica leaf water extract attenuates chlorpyrifos-induced toxicity, rather than simply describing some conventional indices.

(2) Are all plant extracts with antioxidant properties able to alleviate chlorpyrifos-induced toxicity?

(3) lines 29-36, there is a logical confusion, please rewrite.

(4) line 37, “without negative environmental impact”, this viewpoint of the author is too far-fetched and is recommended to be deleted.

(5) It is suggested that “2.4. Plant Material” be moved to the end of “2.1. Reagents”.

(6) line 227, “at a dosage of 16 mg/kg daily”, what is the basis for this intervention dose? This concentration does not correspond to reality, and under normal circumstances it is difficult for chlorpyrifos to reach such high concentrations in the human body. 

(7) lines 319, Table 1 why dose the ethanol-soluble extractive value decreases the ethanol-soluble extractive value? The authors need to give a reasonable explanation.

(8) Tables 1 and 2 need to add the significance analysis.

(9) In animal experiments, the authors need added a positive control group capable of mitigating chlorpyrifos toxicity.

(10) lines 409-411, “a higher body weight in rats from the group receiving low and middle doses of the extracts combined with the insecticide compared to that of the control group on the first day”, is this a good result? why?

(11) There is no significant difference between any of the groups of data in Table 6?

Author Response

This study aimed to investigate the amelioration effect of Litsea martabanica leaf water extract on chlorpyrifos-induced toxicity. The purpose of this study is interesting, and this is a topic of interest to readers, but this study fails to explain the molecular mechanisms involved. Therefore, I recommend that this manuscript undergoes a major revision before judging its suitability for publication in the journal Foods.

(1) The authors need to deepen their research on the molecular mechanisms by which Litsea martabanica leaf water extract attenuates chlorpyrifos-induced toxicity, rather than simply describing some conventional indices.

Response: The research team has collected information on the use of this medicinal plant, a traditional practice passed down for at least two generations. Farmers in the highlands who used this herb by boiling and drinking it during periods of pesticide use found that they remained in good health and did not experience the usual fatigue associated with insecticides or pesticide usage. This research aimed to find scientific evidence to explain these results. Consequently, extracts were selected, prepared, and tested both in vitro and in animals to explain the mechanism and to develop the traditional knowledge into a product for health.

From the results of this experiment, we agree with the reviewer's recommendation that a study at the molecular mechanism level is necessary. In further study, extracts in various fractions have been prepared for in vitro and animal testing. These experiments will allow for a detailed explanation of the mechanisms involved. It is expected that an in-depth study of the mechanism will be conducted in future research.

(2) Are all plant extracts with antioxidant properties able to alleviate chlorpyrifos-induced toxicity?

Response: From our experiments, we found that plants containing the following compounds—terpenes, phenolics, and flavonoids—are likely to have antioxidant effects as well as mitigate chlorpyrifos-induced toxicity. However, the extent of their effect depends on the concentration of active ingredients in each type of plant.  

(3) lines 29-36, there is a logical confusion, please rewrite.

Response: We have rewritten this section to make it more comprehensible.

(4) line 37, “without negative environmental impact”, this viewpoint of the author is too far-fetched and is recommended to be deleted.

Response: We have considered deleting this word.

(5) It is suggested that “2.4. Plant Material” be moved to the end of “2.1. Reagents”.

Response: As appropriate, "2.4. Plant Material" has been moved to the end of "2.1. Reagents." (Line 100)

(6) line 227, “at a dosage of 16 mg/kg daily”, what is the basis for this intervention dose? This concentration does not correspond to reality, and under normal circumstances, it is difficult for chlorpyrifos to reach such high concentrations in the human body. 

Response: A concentration of 16 mg/kg in laboratory animals (animal equivalent dose) is compared to a concentration of 160 mg/day for a person weighing 60 kg or approximately 2.67 mg/kg body weight (human equivalent dose).

In Thailand, the label instructions recommend using 10–50 grams of chlorpyrifos per rai (1 rai = 3.95 acre) of plants. Normally, farmers treat more than one rai per day. Therefore, considering the potential exposure through ingestion, eyes, skin, and inhalation, exposure to chemicals at a level of 160 mg/day is possible.

(7) lines 319, “Table 1” why dose the ethanol-soluble extractive value decreases the ethanol-soluble extractive value? The authors need to give a reasonable explanation.

Response: The reviewer's intended statement is that the extractive value of ethanol is less than the extractive value of water. If so, the reason is that different solvents have different characteristics. Water has a stronger polarity than ethanol, which causes a difference in their solubility. According to the researcher's assumption, this particular herb has substances with a high water-soluble extractive value and polarity.

(8) Tables 1 and 2 need to add the significance analysis.

Response: Because Table 1 is a report on quality control of herbal raw materials in accordance with the Thai herbal pharmacopoeia, no comparisons are made. In contrast, Table 2 has been statistically changed to be comparable to gallic acids. (Line 286)

(9) In animal experiments, the authors need added a positive control group capable of mitigating chlorpyrifos toxicity.

Response: The use of a positive control involves a group that receives substances known to effectively reduce chlorpyrifos toxicity, which may include active ingredients from plants or standard medicines for comparison with the extracts. However, there is currently no standard medicine for treating liver fibrosis or detoxification in the liver. Thus, this research report does not include a positive control group treated with such medicine.

(10) lines 409-411, “a higher body weight in rats from the group receiving low and middle doses of the extracts combined with the insecticide compared to that of the control group on the first day”, is this a good result? why? 

 Response: Therefore, if any rat brought into this research weighs more than 20 grams different from the others, the researcher would exclude it from the study. In this research, the weights of the rats varied but remained within acceptable criteria. Although the weight of the rats in each group was statistically different, the difference was only slight, less than 20 grams.

(11) There is no significant difference between any of the groups of data in Table 6?

Response: We have re-checked and found that when testing the statistical values of all parameters in Table 6, no significant differences were found (All values indicated a non-normal distribution, the Kruskal–Wallis nonparametric ANOVA test followed by Dunn's test was applied.).

Round 2

Reviewer 3 Report

Comments and Suggestions for Authors

ok